# AVSU-Bench and VSpeech-R1: A Dataset and MLLM for Audio-Visual Speech Understanding

## Abstract

Audio-visual speech processing leverages visual cues (e.g., lip movements) to enhance speech robustness in noisy environments. However, current research is heavily focused on Audio-Visual Speech Recognition (AVSR), which primarily addresses the surface-level task of transcription, overlooking the need for deeper semantic understanding under challenging auditory conditions. To bridge this gap, we introduce **Audio-Visual Speech Understanding (AVSU)**, a new task that aims to comprehend semantics and context beyond mere transcription. To support AVSU, we build **AVSU-Bench**, a large-scale dataset with 50k question-answer pairs aligned with audio-visual speech videos. We further propose **VSpeech-R1**, the first-ever end-to-end multimodal large language model tailored for AVSU. A key component of this model is VSpeech-CoT, a structured Chain-of-Thought reasoning framework enabled by a training strategy combining supervised cold-starting and reinforcement learning. Extensive evaluations on AVSU-Bench demonstrate that our end-to-end framework consistently outperforms traditional cascaded pipelines. Specifically, VSpeech-R1 achieves a BERTScore of 92.43%, an absolute improvement of 2.33% over the best cascaded baseline.

## 1 Introduction

In the world of ubiquitous noise, audio-visual speech processing has emerged as a vital approach for enhancing human-machine communication. By integrating acoustic signals with visual cues such as lip movements, audio-visual speech recognition (AVSR) (Son Chung et al., 2017) systems improve robustness and transcription accuracy than traditional audio-only automatic speech recognition (ASR), particularly under noisy conditions. Recent efforts in AVSR (Shi et al., 2022a; Ma et al., 2023; Haliassos et al., 2024; Djilali et al., 2024; Cappellazzo et al., 2024) have primarily focused on improving transcription accuracy from the multimodal inputs. However, this exclusive focus on transcription constrains the applicability of audio-visual speech processing in many real-world scenarios, where successful interaction depends not only on recognizing what is said but also on understanding what is meant. For instance, in applications such as vibe coding or autonomous driving, systems must interpret user intent rather than simply produce verbatim textual transcriptions. This gap highlights a key limitation in current audio-visual speech processing research and underscores the need for next-generation systems capable of semantic understanding in challenging acoustic environments.

To address the limitations of AVSR, we propose a new task: **Audio-Visual Speech Understanding (AVSU)**. Unlike AVSR, which focuses on transcription, AVSU aims to comprehend the speaker's intent in challenging auditory conditions by leveraging both speech and lip movement cues. To facilitate research in this new direction, we present **AVSU-Bench**, a large-scale dataset comprising over 50k question-answer pairs aligned with audio-visual speech videos, to support the training and evaluation of speech understanding models in complex, noisy environments. The proposed AVSU task presents unique challenges that go beyond those of AVSR. First, semantic understanding is inherently more ambiguous and context-dependent than transcription, requiring reasoning over incomplete or corrupted multimodal cues. While AVSR relies on word-level alignment, AVSU demands a deeper fusion of acoustic and visual information to infer speaker intent. Second, the cascading approach, where the output of an AVSR model feeds into a language understanding module,

commonly the Large Language Models (LLMs), struggles in noisy environments due to error propagation. Recognition errors in the first stage can degrade downstream understanding, and modality interactions are still under exploration in such pipeline architectures. Finally, modeling intent understanding requires not only perceptual fidelity but also compositional reasoning and contextual grounding. These challenges necessitate a rethinking of the dataset, architecture, and evaluation protocols, specifically tailored for AVSU.

Building on AVSU-Bench, we propose **VSpeech-R1**, the first unified framework to simultaneously perform audio-visual speech recognition and semantic understanding in a single end-to-end model. This integrated design is fundamental to our approach, as recognition and understanding are inherently intertwined in AVSU. A key component of VSpeech-R1 is **VSpeech-CoT**, a multimodal chain-of-thought (CoT) reasoning mechanism designed to perform inference by jointly leveraging audio and visual inputs. Specifically, VSpeech-CoT aligns acoustic and lip movement cues temporally and integrates them into a four-stage structured reasoning process, promoting context-aware and interpretable intent predictions. This design is particularly effective in noisy auditory environments, where lip movement cues play a crucial role in maintaining robust semantic understanding. To fully leverage the reasoning potential of VSpeech-CoT, we employ a two-phase training strategy. We first use cold-start fine-tuning to initialize the model's reasoning pathways. Subsequently, we apply reinforcement learning to refine its decision-making capabilities with format and accuracy rewards, encouraging coherent and interpretable reasoning chains. This hybrid training strategy enables the model to rapidly acquire structured reasoning capabilities from a small set of annotated samples, and then stabilize and generalize these capabilities through large-scale unlabeled data.

Through extensive experiments, we demonstrate that even a strong cascaded system combining a powerful AVSR model with a language understanding module, still exhibits significant performance limitations on the AVSU task. These pipelines suffer from downstream error propagation and fail to capture the tight coupling of perception and reasoning needed for robust intent understanding. In contrast, our end-to-end baseline performs better, highlighting the advantage of jointly modeling audio-visual-language signals for semantic understanding. Furthermore, our proposed VSpeech-R1, enhanced by the VSpeech-CoT reasoning paradigm, delivers additional performance gains through structured multimodal CoT reasoning. These results not only validate the effectiveness of a unified modeling approach for AVSU, but also underscore the potential and challenges of this task in real-world auditory conditions.

To summarize, our key contributions are as follows:

- We formulate a new task, **Audio-Visual Speech Understanding (AVSU)**, aims at achieving robust semantic understanding using both speech signals and lip movement cues.

- To support the AVSU task, we introduce **AVSU-Bench**, a large-scale dataset designed to train and evaluate models under realistic and challenging auditory conditions.

- For the AVSU task, we propose **VSpeech-R1**, the first end-to-end framework. To further enhance multimodal reasoning in adverse auditory scenarios, we introduce **VSpeech-CoT**, a structured reasoning paradigm with audio-visual context awareness.

- We conduct extensive evaluations showing that our unified, end-to-end approach consistently outperforms strong cascaded baselines that decouple audio-visual speech recognition and semantic understanding. These results highlight the effectiveness of our proposed task, dataset, and method for robust audio-visual speech understanding.

## 2 RELATED WORK

### 2.1 AUDIO-VISUAL SPEECH PROCESSING MEETS LLMS

Audio-Visual Speech Processing (AVSP) leverages the synergy between audio signals and visual cues, particularly lip movements, to enhance speech processing in challenging auditory environments (Summerfield, 1979; Ivanko et al., 2023). Within this domain, AVSR has emerged as a central task, attracting increasing attention due to its potential for robustness in real-world noisy conditions. Landmark foundational models, such as AV-HuBERT (Shi et al., 2022b), have demonstrated that self-supervised learning can effectively capture powerful cross-modal speech representations, while methods like Auto-ASVR (Ma et al., 2023) demonstrate that scaling up datasets can significantly

improve model robustness and accuracy. More recently, the integration of large pre-trained models into AVSR has attracted growing and sustained research interest. For instance, Whisper-Flamingo (Rouditchenko et al., 2024) introduces lip movement features into the speech foundation model Whisper (Radford et al., 2023), thereby producing a notable improvement in AVSR performance. Notably, Multimodal Large Language Models (MLLMs) like LLaMA-AVSR (Cappellazzo et al., 2024) and LLaMA-SMoP (Cappellazzo et al., 2025) align speech and lip movement features with text embeddings, leveraging the power of LLMs to achieve state-of-the-art results in AVSR. Despite these advancements, current research in AVSP has primarily concentrated on improving robust speech recognition or transcription through ASVR. However, human-machine communication in real-world settings involves not only speech recognition but also comprehensive spoken language understanding (Marslen-Wilson & Tyler, 1980; Wang et al., 2005; Serdyuk et al., 2018; Qin et al., 2021), which needs deeper semantic and context understanding. In this work, we introduce the concept of Audio-Visual Speech Understanding (AVSU), aiming to propel the AVSP field beyond superficial multimodal semantic recognition and toward more nuanced semantic comprehension under challenging auditory conditions.

## 2.2 MULTIMODAL CHAIN-OF-THOUGHT REASONING

CoT reasoning (Wei et al., 2022; Kojima et al., 2022; Chen et al., 2025) emulates human-like problem-solving by breaking down complex problems into smaller and tractable sub-components, thereby enabling stepwise solution construction. The intermediate reasoning steps (i.e., rationale), explicitly articulate the logical progression toward a final conclusion, enhancing transparency and interpretability. Extending this paradigm, Multimodal Chain-of-Thought (MCoT) reasoning (Wang et al., 2025b) integrates multimodal signals (e.g., images, videos, audio) into the CoT process. This augmentation expands the reasoning framework's capacities, enabling it to address more complex and diverse scenarios with greater efficacy. Within MCoT, structured reasoning has emerged as a key approach. It seeks to provide better control and interpretability of reasoning processes, thereby improving the accuracy of the reasoning results. However, a critical bottleneck lies in the CoT labeling process, which typically requires substantial human effort and resources, limiting the scalability of structured reasoning approaches. Recently, inspired by the success of DeepSeek-R1 (Guo et al., 2025), numerous studies (Meng et al., 2025; Deng et al., 2025; Shen et al., 2025; Xing et al., 2025) have leveraged reinforcement learning (RL) to enable LLMs to autonomously explore and sustain long CoT reasoning. By aligning the stage labels within the model outputs (e.g., $< think > ... < /think >$) with reward signals that prioritize logical coherence and final results accuracy, reinforcement learning significantly reduce the resource demands for executing structured reasoning workflows, thereby enhancing their practical applicability in real-world scenarios. Inspired by these techniques, we propose a four-stage VSpeech-CoT framework to enable structured multimodal reasoning in AVSU. To bootstrap this capability, we begin with supervised training on on a small set of ∼2.5K samples, each carefully annotated with explicit rationales to ensure reliable reasoning pathways. Building upon this foundation, we then scale up CoT training using reinforcement learning, allowing the model to perform behaviorally consistent and context-aware reasoning across both speech and lip modalities.

## 3 AVSU TASK AND DATASET

As shown in Figure 1, the proposed AVSU task aims to develop a multimodal system that is capable of interpreting and understanding spoken language through both audio (i.e., speech) and visual (i.e., lip movement) modalities, while also considering a text-based question. Also, to comprehensively evaluate the quality of generated answers, we adopt BERTScore (Zhang et al., 2019) ($S_{\text{BERT}}$) as our primary metric to capture semantic similarity through contextual embeddings. In the following sections, we present a detailed description of the AVSU-Bench dataset construction and verification, which serves as the dataset foundation for training and evaluating models on this newly proposed task.

### 3.1 DATASET CONSTRUCTION

Figure 2 illustrates the construction pipeline for the AVSU-Bench dataset. In specific, we use speech transcriptions as contextual input to prompt a text-only LLM, GLM-4-Flash, to generate question-

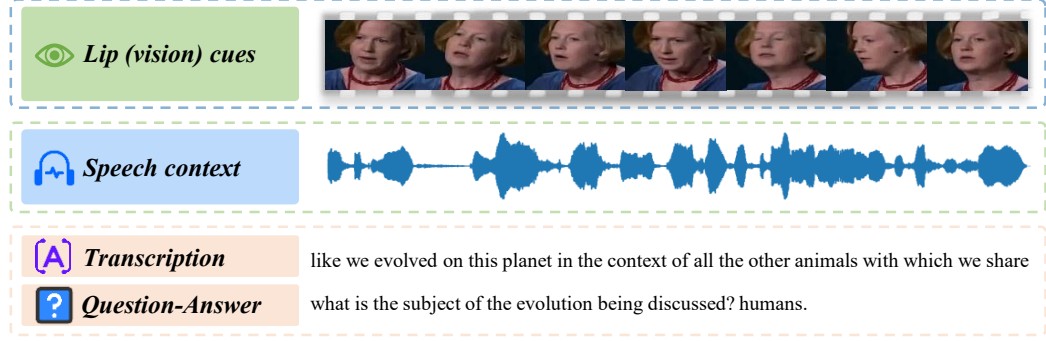

Figure 1: Example of the AVSU-Bench dataset format. The proposed AVSU task is designed to integrate both lip movements and speech context to enable robust and accurate speech understanding in complex auditory environments.

Table 1: Comparison between AVSR and AVSU datasets. AVSU-Bench offers large-scale and high-quality audio-visual data curated from the subset of LRS3 (TED talks), specifically designed to facilitate research in multimodal speech understanding. [†]: AVSU-Bench includes 49,545 training samples and 2,539 test samples.

| Dataset | Task | Source | #Samples | #Hours |
|---|---|---|---|---|
| GRID (Cooke et al., 2006) | Recognition | N/A | 33,000 | 27.5 |
| MODALITY (Czyzewski et al., 2017) | Recognition | N/A | 5,880 | 31 |
| LRW (Chung & Zisserman, 2017) | Recognition | BBC | 1,000 | 173 |
| LRS2 (Son Chung et al., 2017) | Recognition | BBC | 144,243 | 224.5 |
| LRS3 (Afouras et al., 2018) | Recognition | YouTube | 152,452 | 438 |
| AVSU-Bench (ours, 2025) | **Understanding** | YouTube | 52,084[†] | 104 |

answer pairs based on the given context. To ensure training stability, we first filter out videos that are either too short or too long. To improve the quality and relevance of the generated question-answer pairs, we design carefully engineered prompts that contain multiple *context-question-answer* demonstrations[1]. This method enhances the overall data quality while significantly reduces the effort required for subsequent manual verification. Using this pipeline, we initially generate ∼100k question-answer pairs for the training split and ∼10k for the test split. A notable feature of our dataset, as shown in the middle of Figure 2, is that we also extract ∼10k samples from the training set to construct reasoning paths from questions to answers, which we refer to as *rationales*.

## 3.2 DATASET VERIFICATION

To ensure the quality and reliability of AVSU-Bench, we implemented a rigorous validation protocol. For the training set, given its large scale, we first perform an initial validation using GLM-4-Flash (denoted as "junior check" in Figure 2), followed by a more thorough review using GPT-4o (denoted as "senior check"). For the subset of training samples augmented with CoT rationales, we directly apply the senior check with GPT-4o to verify the correctness and coherence of the reasoning paths. For the test set, we adopt a similar multi-stage validation strategy, but with a crucial difference: instead of relying on LLMs, we perform a manual review to ensure maximum accuracy and quality. We also control the ratio of question types (e.g., yes/no vs. open-ended) to maintain a balanced evaluation.. Finally, as summarized in Table 5, we introduce AVSU-Bench, a dataset specifically designed for audio-visual speech understanding.

---

[1]During the dataset construction, we use speech transcriptions to simulate spoken context within the text-based LLM.

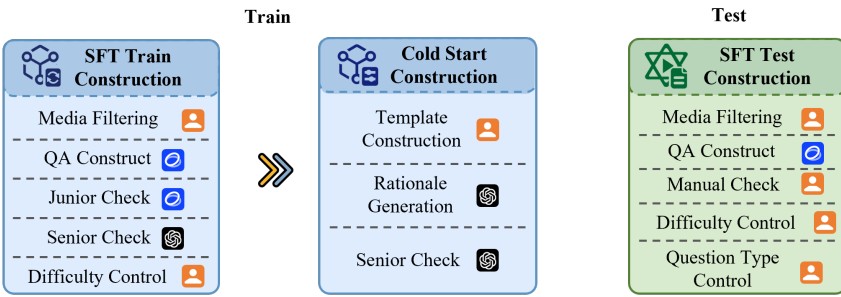

Figure 2: Overview of AVSU-Bench construction pipeline. The annotation process incorporates three-level quality assurance, including junior check, senior check, and manual check to balance the annotation efficiency and reliability.

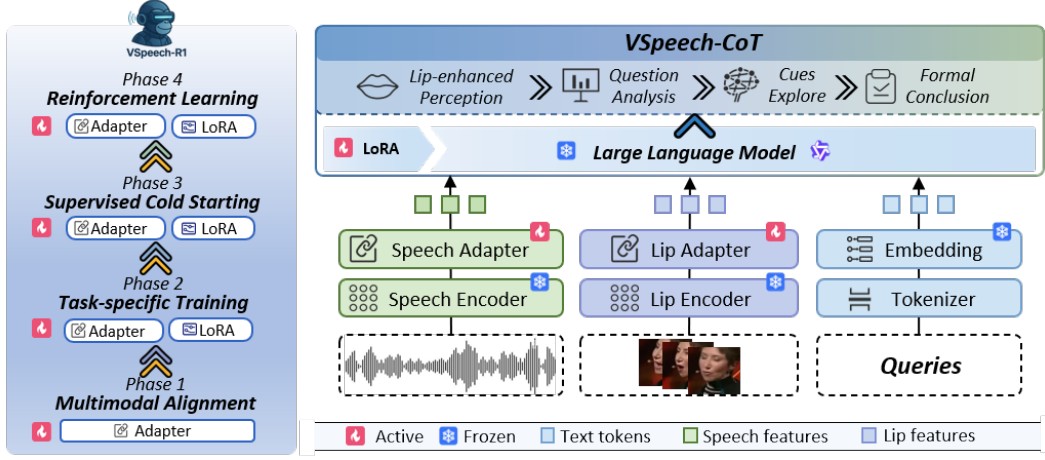

Figure 3: VSpeech-R1 architecture overview (right) and training strategies (left) with activated modules. VSpeech-R1 consists of separate encoders for speech and vision (e.g., lip) modalities. The speech and vision tokens are aligned with text embedding space via a single-layer linear projection, while the base text LLM is optimized through LoRA for parameter-efficient training.

## 4 VSPEECH-COT AND VSPEECH-R1

### 4.1 OVERALL ARCHITECTURE

Figure 3 illustrates the overall architecture of the proposed VSpeech-R1, leveraging the reasoning capabilities of an LLM for audio-visual speech understanding.

**Multimodal Encoder:** We use Whisper (Radford et al., 2023) and AV-HuBERT (Shi et al., 2022b) as the speech and vision encoders for our VSpeech-R1, respectively. Both encoders are kept frozen to leverage their powerful pre-trained representations. We further use a single-layer projection as a modality adapter to align these multimodal features with the LLM's text embedding space.

**Language Decoder:** We use Qwen3-8b (Yang et al., 2025) as the base LLM for language decoding. We train the base LLM using LoRA, which allows us to adapt its behavior for our specific task while keeping the base LLM itself frozen. Textual information and prompts are processed together by a tokenizer before being input into the LLM. Multimodal features are concatenated with text modality and fed into LLM for the next token prediction.

**VSpeech-CoT Framework:** To advance AVSU with robust multimodal reasoning, we propose a novel four-stage VSpeech-CoT reasoning framework, which serves as a core reasoning mechanism

of VSpeech-R1. As shown at the top of Figure 3, VSpeech-CoT systematically progresses through four key stages: (1) *Lip-enhanced Perception* for lip-enhanced robust speech context perception; (2) *Question Analysis* for identifying the reasoning objectives; (3) *Cues Exploration* for integrating cross-modal evidence for final answer derivation; and (4) *Formal Conclusion* for generating the final answer. To operationalize this framework, we collect 2.5K VSpeech-CoT samples for supervised cold-starting, which activates the model's initial capability to follow the reasoning structure of VSpeech-CoT. We then scale up the training capacity with reinforcement learning, which enables the model to perform consistent and context-aware reasoning across both speech and lip modalities.

## 4.2 MODEL TRAINING

### 4.2.1 MULTIMODAL ALIGNMENT

Given that AVSR aims to learn semantically aligned representations across modalities, we formulate this task as aligning speech and lip movements with the textual modality to enable effective multimodal integration. To this end, we use the LRS3 dataset, which provides over 400 hours of speech-video data. Speech and lip features are independently projected into the LLM's embedding space using modality-specific single-layer linear projectors. We then perform a joint tri-modal alignment training to enhance cross-modal consistency. Notably, directly align tri-modal features led to a slow alignment process and suboptimal performance in our practical experience.

### 4.2.2 TASK-SPECIFIC TRAINING

Once the modalities are aligned, we bootstrap the model's core audio-visual speech understanding capability with the proposed AVSU-Bench dataset. At this stage, we employ LoRA-based parameter-efficient training on both the base LLM and the multimodal adapters, enabling task-specific adaptation for AVSU while preserving cross-modal consistency. The overall training procedure, including data augmentation, follows a similar setup used in the multimodal alignment phase.

### 4.2.3 SUPERVISED COLD-STARTING

While task-specific training enables the model to acquire fundamental audio-visual speech understanding capabilities, it falls short of imparting the structured reasoning skills required by our VSpeech-CoT framework. To bridge this gap, we adopt a supervised cold-starting strategy by training the model on a curated set of VSpeech-CoT samples annotated with explicit reasoning rationales, thereby bootstrapping its ability to perform structured CoT reasoning.

### 4.2.4 REINFORCEMENT LEARNING

Reinforcement learning forms the core training paradigm of VSpeech-R1, enabling the model to efficiently and scalably learn structured reasoning patterns within the VSpeech-CoT framework. To implement this, we employ Group Relative Policy Optimization (GRPO) (Guo et al., 2025) as the reinforcement learning strategy and design a set of simple yet effective rule-based rewards to guide the model in generating reasoning chains that align with the VSpeech-CoT structure.

Specifically, we define five hierarchical format rewards to ensure both behavioral consistency and adherence to the CoT format: (a) *Rationale Existence Reward* encourages the generation of a rationale for each inference; (b) *Stage Completeness Reward* ensures the full completion of each reasoning stage; (c) *Stage Uniqueness Reward* promotes uniqueness and avoids redundancy in reasoning; (d) *Stage Consistency Reward* ensures the model covers the same reasoning stages prescribed by VSpeech-CoT; and (e) *Stage Ordering Reward* enforces the correct order of reasoning stages. Additionally, these format rewards are combined with an *Answer Accuracy Reward*, directly optimizing the correctness of the final answer. This comprehensive reward structure enables the model to improve both its structured reasoning process and the accuracy of its output. The aforementioned rule-based rewards impose progressively finer constraints on the rationale construction within VSpeech-CoT. Compared with using supervised cold-starting alone, VSpeech-CoT enhanced with reinforcement learning for scalable reasoning pattern training achieves improved reasoning completeness and answer accuracy, as discussed in Sections 5.1 and 5.2. In addition, the effects of the format and accuracy rewards on overall performance are discussed in Section 5.3.

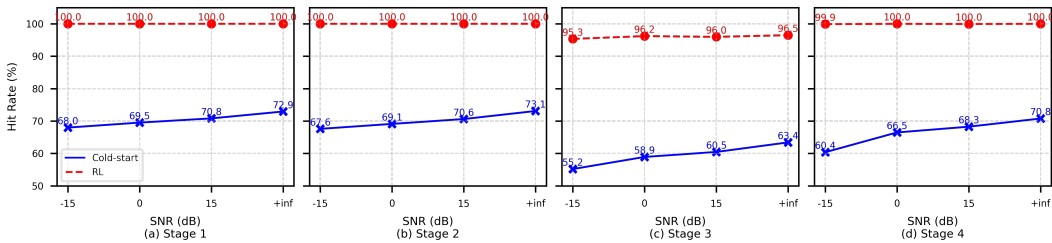

Figure 4: Hit rate comparison across different reasoning stages of VSpeech-CoT, evaluating the impact of reinforcement learning on the reasoning completeness. Subfigures (a) to (d) correspond to the four reasoning stages: (a) lip-enhanced perception, (b) question analysis, (c) cues exploration, and (d) formal conclusion.

Table 2: Results on AVSU-Bench under varying SNR levels, evaluated with $S_{\text{BERT}}$ (higher is better). "SFT" denotes supervised fine-tuning, and "+Inf" indicates the clean audio setting. All experiments use the Qwen3-8B backbone.

| Pattern | Model | +Inf | +15 | +5 | 0 | -5 | -15 |
|---------|-------|------|-----|----|----|----|----|
| Cascaded | Whisper-Flamingo + Qwen (w/o SFT) | 79.90 | 79.85 | 79.72 | 79.54 | 78.99 | 75.30 |
| Cascaded | Whisper-Flamingo + Qwen (w/ SFT) | 90.10 | 89.78 | 89.60 | 88.91 | 87.41 | 85.25 |
| End-to-End | VSpeech-R1 (SFT) | 90.41 | 90.12 | 89.60 | 89.03 | 87.96 | 86.11 |
| End-to-End | VSpeech-R1 (Cold-start) | 91.92 | 91.45 | 90.96 | 90.42 | 89.04 | 87.13 |
| **End-to-End** | **VSpeech-R1 (RL)** | **92.43** | **91.94** | **91.53** | **91.17** | **90.01** | **87.98** |

## 5 EXPERIMENTS

### 5.1 VSPEECH-CoT COMPLETENESS

As illustrated in Figure 4, we evaluate the effectiveness of cold-starting and reinforcement learning on the hit rate, which computes the rate that the model successfully completes each single stage of the VSpeech-CoT reasoning process. After cold-starting, the model exhibits a preliminary ability to follow the structured reasoning stages of VSpeech-CoT. However, due to the limited scale of annotated samples in this phase, the consistency of output format and reasoning behavior remains suboptimal. Specifically, following cold-start initialization, the hit rates for Stage 1 and Stage 2 reach approximately 70%, while Stage 4 achieves a hit rate between 60% and 70%. In contrast, Stage 3 lags behind, with a hit rate of only around 50% to 60% across different Signal-to-Noise Ratio (SNR) conditions. Moreover, since rationale construction and verification in the cold-start phase rely on automated processes, there remains a risk of introducing incorrect or noisy supervision.

To mitigate the limitations of cold-start initialization and improve the completion rate of reasoning under the VSpeech-CoT framework, we further apply reinforcement learning. As shown in Figure 4, after the reinforcement learning phase, the model achieves near-perfect performance across all reasoning stages, with hit rates approaching or reaching 100%. In addition, the reasoning process demonstrates strong robustness under varying SNR conditions. In contrast, the model trained solely with cold-start initialization is significantly more vulnerable to noise, exhibiting a 5%–10% hit rate drop under extreme noise settings.

### 5.2 QUANTITATIVE RESULTS ON AVSU-BENCH

As shown in Table 2, we conduct comprehensive evaluations on AVSU-Bench under varying SNR conditions, ranging from clean audio ("+Inf") to severely degraded signals (-15dB). We report results using the $S_{\text{BERT}}$ metric, which measures semantic alignment between generated and reference responses, with higher scores indicating better understanding.

We compare two system paradigms: (1) *Cascaded systems*, which decouple speech recognition and language understanding, and (2) *End-to-End systems*, which jointly model audio-to-text reasoning within a unified architecture.

The cascaded baseline, Whisper-Flamingo + Qwen (w/o SFT), exhibits consistently poor performance across all SNR levels. It achieves only 79–80% $S_{\text{BERT}}$ under clean and moderate conditions, with performance further degrading to 75.30% at -15dB. These results indicate that, such a cascaded system struggles to effectively leverage contextual and cross-modal cues, especially under noisy conditions. Through supervised fine-tuning, the cascaded baseline (i.e., Whisper-Flamingo + Qwen (w/ SFT)) demonstrates substantial gains, raising $S_{\text{BERT}}$ from 79.90% to 90.10%. This improvement underscores the critical role of our proposed **AVSU-Bench** in enhancing the robustness and effectiveness, even when applied to cascaded pipelines.

In contrast, our proposed end-to-end VSpeech-R1 model consistently surpasses all cascaded counterparts across every SNR conditions. Both its SFT and Cold-start variants exhibit strong robustness to noise, even without reinforcement learning. Remarkably, VSpeech-R1 (Cold-start) attains 91.92% on clean inputs and sustains performance above 87.0% at -15dB, indicating that the model effectively acquires noise-invariant and structured reasoning capabilities. The best overall performance is achieved by VSpeech-R1 (RL), which leverages reinforcement learning to further align model inference with downstream reasoning objectives. This variant reaches a peak $S_{\text{BERT}}$ of 92.43% in clean conditions and retains 87.98% under severe noise, consistently outperforming all baselines. These results highlight the critical role of structured, CoT-driven reasoning in enhancing semantic robustness and maintaining consistency under adverse auditory conditions.

## 5.3 ABLATION STUDIES

As shown in Table 3, to better understand the contribution of each component in our framework, we conduct an ablation study with different model configurations. The settings vary across three key dimensions: supervised fine-tuning (SFT), CoT cold-starting (Cold-start), and reinforcement learning (RL) with two reward signals (i.e., format consistency $\mathcal{R}_{\text{fmt}}$ and result accuracy $\mathcal{R}_{\text{acc}}$).

Table 3: Ablation study of VSpeech-R1.

| Setting | SFT | Cold-start | RL | | $S_{\text{BERT}}$ |
| --- | --- | --- | --- | --- | --- |
| | | | $\mathcal{R}_{\text{fmt}}$ | $\mathcal{R}_{\text{acc}}$ | |
| 1 | ✓ | ✗ | ✗ | ✗ | 90.41 |
| 2 | ✓ | ✓ | ✗ | ✗ | 91.92 |
| 3 | ✓ | ✓ | ✓ | ✓ | 92.43 |
| 4 | ✓ | ✗ | ✓ | ✓ | 90.64 |
| 5 | ✓ | ✓ | ✓ | ✗ | 92.26 |
| 6 | ✓ | ✓ | ✗ | ✓ | 91.86 |

**Effectiveness of Cold-start:** The comparison between Setting 1 and Setting 2 highlights the effectiveness of cold-start initialization. While SFT alone achieves 90.41% $S_{\text{BERT}}$, introducing cold-start (Setting 2) further improves performance to 91.92%. This demonstrates that a pre-fixed and well-initialized reasoning pathway can enhance reasoning capability, even without reinforcement learning.

**Effectiveness of RL:** Setting 3, which integrates the complete reinforcement learning framework and jointly optimizes both the format reward ($\mathcal{R}_{\text{fmt}}$) and task accuracy reward ($\mathcal{R}_{\text{acc}}$), achieves the highest overall performance with an $S_{\text{BERT}}$ score of 92.43%. This demonstrates the effectiveness of rule-based reward-driven optimization in enhancing audio-visual speech understanding.

**Ablating Individual Rewards:** To better understand the contribution of each reward signal, we further disentangled their effects. In Setting 5, removing $\mathcal{R}_{\text{acc}}$ results in a slight performance decline to 92.26%. However, in Setting 6, removing $\mathcal{R}_{\text{fmt}}$ results in a larger drop to 91.86%, highlighting the importance of well-defined format reward in enhancing AVSU with reinforcement learning.

**Necessity of Cold-Start in RL:** To assess whether reinforcement learning benefits from cold-start initialization, we compare Setting 4 (RL w/o cold-start) with Setting 3 (RL w/ cold-start). The performance in Setting 4 drops to 90.64%, almost on par with SFT-only baseline. This indicates that without proper initialization, reinforcement learning may struggle to find effective reasoning pathways, underscoring the necessity of our cold-starting phase.

## 5.4 QUALITATIVE RESULTS

As shown in Table 4, we present a qualitative evaluation on AVSU-Bench under challenging speech conditions. Sample (a) highlights the effectiveness of different baselines, covering both cascaded and end-to-end paradigms. Sample (b) illustrates the limitations of cascaded baselines, which fail to provide correct responses in challenging auditory scenarios. Sample (c) further demonstrates these

Table 4: Qualitative results on the test set of AVSU-Bench under -15 and -5 NSR condition. The phrases marked in red indicate incorrect answers (i.e., text responses), while the phrases marked in green indicate correct answers, in comparison with the ground truth labels marked in orange.

| | Sample (a) | | Sample (b) | | Sample (c) | |
|---|---|---|---|---|---|---|
| Context (speech) | So we decided to go to the mosque. | | That's not an improbable sample. | | Actually studied hormones. | |
| Question-Answer | What is the destination they chose? mosque | | Can this sample be regarded as probable? yes | | What subject was studied? hormones | |
| | -15 NSR | -5 NSR | -15 NSR | -5 NSR | -15 NSR | -5 NSR |
| Cascade (w/o SFT) | they decided ... mosque | they decided ... mosque | invalid content | we cannot guarantee ... | invalid content | telephone |
| Cascade (w/ SFT) | mosque | mosque | no | yes | psychology | music |
| VSpeech-R1 (SFT) | mosque | mosque | it is true | yes | not specified | hormones |
| VSpeech-R1 (Cold-start) | they may ... mosque | mosque | yes | yes | not mentioned | hormones |
| VSpeech-R1 (RL) | mosque | mosque | yes | yes | hormones | hormones |

limitations, while a comparison across different end-to-end variants of our proposed VSpeech-R1 clearly showcases its superior robustness under adverse conditions.

# 6 DISCUSSION AND FUTURE WORKS

**Reasoning Efficiency.** While CoT reasoning enhances response accuracy by encouraging structured and interpretable inference (Wang et al., 2025b; Chen et al., 2025), it may incur higher latency (Feng et al., 2025). VSpeech-R1 integrates VSpeech-CoT to enable chain-based reasoning within the AVSU framework. To balance reasoning performance and computational efficiency, future work could investigate hybrid reasoning strategies (Shang et al., 2024; Jiang et al., 2025). In this setup, the model dynamically adjusts its inference depth based on question complexity, in specific, CoT reasoning is applied to more difficult or ambiguous queries, while direct, single-step inference is used for simpler ones. Furthermore, large scale VSpeech-CoT models may serve as teacher models to distill structured reasoning patterns into smaller and faster student models. This approach preserves reasoning quality while reducing inference overhead.

**Emotional Cues.** While lip movements provide strong phonetic cues for speech recognition, they also carry subtle emotional and affective information. Micro-expressions, facial muscle tension, and speaking dynamics (e.g., pauses, tempo, lip tightness) can offer critical cues for understanding speaker intent, attitude, and context, especially in emotionally charged or ambiguous utterances. Future work can explore integrating visual emotion recognition modules into the AVSU pipeline, enabling the model to infer not only what is being said, but also how it is being said. This integration holds promise for enhancing downstream applications such as affective computing (Wang et al., 2022; Canal et al., 2022; Wang et al., 2023; Amin et al., 2024), persuasive dialogue understanding (Wang et al., 2019; Chen et al., 2021; Jin et al., 2023) and beyond semantic speech (Wang et al., 2025a), ultimately enabling more human-aligned and contextually aware speech understanding.

# 7 CONCLUSION

This work introduces **Audio-Visual Speech Understanding (AVSU)**, a new task that aims to bridge audio-visual speech recognition and semantic understanding by leveraging auditory and visual modalities, either independently (i.e., the cascade pattern) or end-to-end. To facilitate the development of this novel task, we present **AVSU-Bench**, the first dataset for AVSU, which contains ~50k question-answer pairs and over 100 hours of annotated audio-visual speech data. AVSU-Bench includes a manually curated test set of ~2.5k samples, specifically designed to enable a rigorous assessment of model performance in challenging auditory conditions. We further propose **VSpeech-R1**, the first MLLM tailored for AVSU, equipped with **VSpeech-CoT**, a four-stage MCoT reasoning framework for robust AVSU. Our experimental results, both quantitative and qualitative, demonstrate that visual cues from lip movements substantially enhance semantic understanding, particularly in scenarios with severe acoustic degradation. These findings highlight the importance of tightly integrated audio-visual-language modeling and point toward promising directions for advancing audio-visual speech processing in complex real-world settings. Future work could further explore more efficient hybrid reasoning strategies and investigate using emotional cues to facilitate more accurate intent understanding.

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

# A  APPENDIX

## A.1  DATASET STATISTICS

After thorough validation, we finalized approximately 50k training samples and 2.5k test samples. Among them, ∼2.5k instances in the training set are additionally annotated with explicit rationales to facilitate structural reasoning during the cold-starting phase. Table 1 summarizes the statistics of AVSU-Bench alongside other related datasets. Figure 5 presents detailed statistics on the distribution of question formats as well as the average length of videos, contexts, questions, and answers. Notably, For the test set, we manually balanced the distribution of question types to ensure fair and comprehensive evaluation across different semantic understanding conditions. To the best of our knowledge, AVSU-Bench is the first dataset specifically designed to advance robust speech understanding by incorporating lip movements as visual cues.

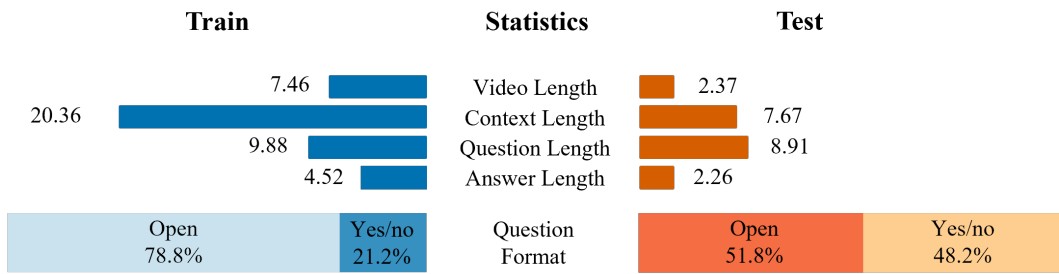

Figure 5: Data distribution of the AVSU-Bench dataset. The duration of videos is reported in seconds, whereas the lengths of questions and answers are quantified by word count.

## A.2  REINFORCEMENT LEARNING REWARD

We design a set of complementary rewards to guide reinforcement learning, each addressing a specific aspect of the reasoning process. In this section, we describe the design and purpose of each reward used in reinforcement learning. The **Rationale Existence Reward** encourages the model to explicitly generate a rationale for each inference, assigning +0.25 when the output begins with <context> and another +0.25 when it ends with </answer>, without imposing further constraints on intermediate reasoning. The **Stage Completeness Reward** enforces the proper closure of reasoning stages by granting +0.25 for each matched start–end pair (<stage> and </stage>), or +0.1 when only a single unmatched marker is present. To promote diversity and discourage redundancy, the **Stage Uniqueness Reward** penalizes repeated stages with −0.05 for each additional occurrence. The **Stage Consistency Reward** aligns reasoning with the VSpeech-CoT framework by imposing a penalty of −0.1 whenever a stage outside the prescribed framework appears, while the **Stage Ordering Reward** further enforces structural coherence by granting +0.333 for each pair of consecutive stages that follows the correct predefined order. Finally, the **Answer Accuracy Reward** directly optimizes the model based on the correctness of its final predicted answer.

## A.3  EXPLORING THE BASELINE ARCHITECTURE

We report the ablation results conducted during the pretraining phase to support the final model architecture choice we used. As shown in Table 5, we explore the performance under different Q-Former settings. The performance is evaluated based on the Word Error Rate (WER). The second and third columns represent the Q-Former configurations for the speech and vision branches, respectively. Specially, "/" in Q-Former configuration represents there is no Q-Former used for cross-modal alignment, hence multimodal features encoded by the encoders are fed into linear projection layer directly without the integration of Q-Former. By comparing settings (1) and (2), we observe that introduce the Q-Former does not affect the ASR performance. In settings (3), (4), and (5), we investigate the effect of using a Q-Former in the vision branch for Visual Speech Recognition (VSR) task. In setting (3), we use a standard vision Q-Former with learned vision-relative queries. In contrast, setting (4) initializes the vision-relative queries with the parameters of the speech-relative

Table 5: Early architectural exploration of Q-Former configurations. "/" denotes no Q-Former is used. While Q-Former can improve inference efficiency by reducing token length, it may lead to performance degradation due to the loss of fine-grained information.

| Setting | Speech Q-Former | Vision Q-Former | ASR (%) ↓ | VSR (%) ↓ |
|---------|-----------------|-----------------|-----------|-----------|
| (1) | / | / | **1.1** | - |
| (2) | speech query | / | **1.1** | **27.1** |
| (3) | speech query | vision query | - | 27.7 |
| (4) | speech query | speech query | - | 28.0 |
| (5) | speech query | speech query (frozen) | - | 29.3 |

queries obtained from the first pretraining stage (i.e., ASR), based on the idea that speech and lip movement share temporally aligned semantic information. In setting (5), we follow the configuration of setting (4) but further freeze the queries, aiming to ensure that both speech and lip features are queried using the same queries. The experimental results reveal that the standard configuration in setting (3), where vision queries are randomly initialized, achieves relatively better performance (27.7% vs. 28.0% vs. 29.3%). Nevertheless, it still underperforms compared to the best result (27.1%) obtained in setting (2), which omits the use of a Q-Former. A plausible explanation is that although lip movements provide temporally synchronized semantic cues aligned with speech, their inherently complex spatiotemporal dynamics, involving both motion and structural facial information, make them highly sensitive to information compression. As a result, applying a Q-Former to reduce or downsample the lip feature may inevitably cause information loss. To mitigate this issue and improve performance, we ultimately abandon the Q-Former structure and directly leverage the raw encoder outputs followed by a single-layer linear projection.

## A.4 PERFORMANCE ON PRETRAINING TASK

Table 6 presents the AVSR performance on the LRS3 dataset, measured in terms of word error rate (WER). Although the model is primarily designed for general audio-visual speech understanding, it attains competitive results in audio-visual speech recognition, achieving a WER of 0.98 compared to 0.95 from specialized AVSR models. It is worth noting that in speech recognition, WER differences below 3% are typically regarded as not statistically significant (Gemini et al., 2024).

Table 6: Comparison of AVSR performance on the LRS3 test set against specialized AVSR models. Bold text indicates the optimal performance, while underlined text represents the suboptimal.

| Models | Year | Encoders (Speech) | Encoders (Lip) | WER% ↓ | | |
|--------|------|-------------------|----------------|--------|-----|------|
| | | | | ASR | VSR | AVSR |
| CM-seq2seq (Ma et al., 2021) | 2021 | Transformer | Transformer | 2.3 | 46.9 | 2.3 |
| AV-HuBERT (Shi et al., 2022b) | 2022 | Transformer | Transformer | 1.3 | 26.9 | - |
| AV-data2vec (Lian et al., 2023) | 2023 | Transformer | Transformer | 1.3 | 28.5 | 1.3 |
| RAVEn (Haliassos et al., 2023) | 2023 | Transformer | Transformer | 1.4 | 24.4 | - |
| BRAVEn (Haliassos et al., 2024) | 2024 | Transformer | Transformer | **1.1** | **20.1** | - |
| Whisper-finetuned (Rouditchenko et al., 2024) | 2024 | Whisper | - | 2.3 | - | - |
| Whisper-Flamingo (Rouditchenko et al., 2024) | 2025 | Whisper | AV-HuBERT | - | - | 1.0 |
| LLaMA-AVSR (Cappellazzo et al., 2024) | 2025 | Whisper | AV-HuBERT | **1.1** | 26.9 | **0.95** |
| VSpeech-R1 (ours) | 2025 | Whisper | AV-HuBERT | **1.1** | 27.0 | 0.98 |

## A.5 EXPERIMENT DETAILS FOR REPRODUCIBILITY

To improve model robustness, we apply modality-specific data augmentation strategies: horizontal flipping and random cropping for visual inputs, and white noise injection, gain adjustment, and pitch shifting for audio inputs. The multimodal alignment process is conducted on $4 \times$ A800 (80GB) GPUs with 1e-5 learning rate, and takes approximately seven days to complete. Task-specific training is conducted on $4 \times$ A800 (80GB) GPUs with 1e-5 learning rate and is completed within three days. The cold-starting is conducted on $4 \times$ A800 (80GB) GPUs with 1e-5 learning rate and is completed within two days. The final reinforcement learning is conducted on $2 \times$ A800 (80GB) GPUs with 5e-6 learning rate and is completed within two weeks.

## A.6 VSPEECH-CoT DEMONSTRATIONS

In this section, we provide representative data samples from our dataset. The goal is to illustrate the structure and format of the data instances used in our experiments. Each sample contains four fields: (i) the raw speech transcription, (ii) a corresponding question, (iii) a rationale that demonstrates the reasoning process based on the given context, and (iv) the final answer.

```
"speech_transcription": "happens is that the research focuses on changing
    them a little bit not too much so they stay",
"question": "how much change is considered acceptable in the research?",
"rationale": "<context>happens is that the research focuses on changing
    them a little bit not too much so they stay</context>
<question_analysis>the question asks for the extent of change deemed
    acceptable in the context of the research, which implies a need to
    identify the parameters of change mentioned.</question_analysis>
<cues_explore>the context indicates that the focus is on changing them a
    little bit, and specifies that the change should not be too much.</
    cues_explore>",
"answer": "not too much."
```

Sample 1

```
"speech_transcription": "but guess how most people find work today",
"question": "what is the primary method used by most people to find work
    today?",
"rationale": "<context>but guess how most people find work today</context
    >
<question_analysis>the question seeks to identify the main method people
    use to find employment, implying that the context should provide
    information about job searching methods.</question_analysis>
<cues_explore>the context hints at a discussion about job searching but
    does not specify any methods or details.</cues_explore>",
"answer": "not specified."
```

Sample 2

```
"speech_transcription": "what i found the most valuable was when she said
    that as she started to get healthy that need to share is gone she
    wants to leave those events behind her and move on with her life",
"question": "what action does she wish to take regarding the past events
    ?",
"rationale": "<context>what i found the most valuable was when she said
    that as she started to get healthy that need to share is gone she
    wants to leave those events behind her and move on with her life</
    context> <question_analysis>the question is focused on the action the
     subject wishes to take concerning past events, indicating a desire
    to move forward.</question_analysis> <cues_explore>the context
    mentions that she wants to leave those events behind her and move on
    with her life, suggesting a clear intention to not dwell on the past
    .</cues_explore>",
"answer": "leave them behind."
```

Sample 3

## A.7 DIFFICULTY CONTROL

During dataset construction, we applied a difficulty control strategy. Specifically, we used the Flesch Reading Ease (FRE) score as a measure of sentence readability and retained only the data with readability scores between 20 and 80. The FRE score is computed based on both the average sentence length and the average number of syllables per word, as shown in Equation 1. A higher FRE score indicates greater readability, while a lower score suggests increased difficulty.

$$\text{FRE} = 206.835 - 1.015 \times \left( \frac{\text{total words}}{\text{total sentences}} \right) - 84.6 \times \left( \frac{\text{total syllables}}{\text{total words}} \right), \qquad (1)$$

where $\frac{\text{total words}}{\text{total sentences}}$ represents the average sentence length, and $\frac{\text{total syllables}}{\text{total words}}$ represents the average syllables per word. The constant term $206.835$ and the coefficients $1.015$ and $84.6$ were empirically determined by Flesch (1948) through statistical analysis of English corpora, ensuring that the resulting scores generally fall within the range of 0 to 100. In this scale, higher scores correspond to easier texts, whereas lower scores are associated with more complex or technical writing.

### A.8 USE OF LLMS

In this work, LLMs were used solely to refine and polish the authors' manually written draft for better readability. No LLMs were involved in idea generation or experimental execution.

