# OpenReview forum: "AVSU-Bench and VSpeech-R1: A Dataset and MLLM for Audio-Visual Speech Understanding"
_ICLR.cc/2026/Conference — ICLR 2026 Conference Withdrawn Submission_

### Official Review · Reviewer_dc29 · 2025-10-28

**Soundness:** 3
**Presentation:** 2
**Contribution:** 2
**Rating:** 2
**Confidence:** 4

**Summary:**

This work introduces AVSU-Bench, an audio-visual speech understanding benchmark that aims to capture semantics and context beyond mere transcription. The benchmark contains over 50K question–answer pairs paired with audio-visual speech videos. The authors also propose VSpeech-R1, a unified end-to-end model capable of performing both audio-visual speech recognition and semantic understanding. The model is trained with a chain-of-thought reasoning mechanism to logically interpret audio-visual signals. Experimental results demonstrate that the proposed training approach effectively enables the model to perform audio-visual understanding, outperforming cascaded models.

**Strengths:**

- Existing audio-visual speech tasks mainly focus on transcription. Introducing a benchmark specifically designed for audio-visual speech understanding fills an important gap and will be valuable to the community.

- The proposed chain-of-thought reasoning approach proves effective beyond naïve supervised fine-tuning, enhancing the model’s ability to logically reason about audio-visual speech.

**Weaknesses:**

- In L261, is the lip encoder truly necessary? The role of lip features and the specific information they provide are not clearly described. The ablation study should support the effectiveness of lip features for this task.

- In Figure 5, the overall length of the test set appears much shorter than that of the training data, which may make the benchmark overly easy, given that accuracy already exceeds 90% in Table 2.

- The experimental setup and details are not clearly explained:
  - What is the source dataset used to construct AVSU-Bench?
  - What is the BERT evaluation metric, and how is it measured?
  - How is the answer parsed? For instance, in sample 1 of Section A.6, would the answer “a little” be considered incorrect?

- In L384, it is expected that training on the AVSU-Bench training set improves performance on its test set. How does the model generalize to other benchmarks?

- In Table 2, the reviewer questions whether the degradation in the cascaded system is primarily due to noise. Both the proposed model and the cascaded model show similar degradation (around 4–5%) when moving from clean to noisy (-15 dB) audio. Moreover, the SFT-trained cascaded model performs comparably to the proposed model. This suggests that training on the proposed dataset, rather than the noise condition, primarily determines performance. Additional analysis is needed to support the claim that noise levels significantly affect performance.
Furthermore, output answer formats may influence results — existing LLMs tend to produce long answers, while the proposed dataset favors short answers. This mismatch could cause the cascaded model to score lower on the S-BERT metric. Would training the cascaded Qwen model with RL improve performance?

**Questions:**

### Questions and Suggested Experiments
- Conduct an ablation study to evaluate the usefulness of the lip encoder and clarify its contribution.
- Explore training the cascaded model with RL to assess potential improvements.
- Provide generalization results on other benchmarks to demonstrate the effectiveness of the proposed training method.
- Analyze the relationship between audio noisiness and model performance in more detail.
- Compare the proposed model with existing ASR/AVSR models to assess robustness under noisy conditions.
- Add experiments showing that the cascaded model is more vulnerable to noise than the proposed model.

### Minor Questions and Suggestions
- L107: “ASVR” seems to be a typo.
- L212: “Table 5” should be “Figure 5.”
- L287: Clarify the difference between joint tri-modal alignment and direct bi-modal alignment. What does it mean that the process “slows down the alignment”?

---

### Official Review · Reviewer_XZ1d · 2025-10-29

**Soundness:** 3
**Presentation:** 3
**Contribution:** 2
**Rating:** 4
**Confidence:** 4

**Summary:**

The paper introduces a new task called Audio-Visual Speech Understanding (AVSU), which focuses on achieving robust semantic comprehension by integrating both speech and lip movement information. To support this task, the authors introduce AVSU-Bench, a large-scale dataset designed for training and evaluation under realistic noisy conditions. They further propose VSpeech-R1, an end-to-end framework for AVSU, and VSpeech-CoT, a structured reasoning paradigm that enhances multimodal understanding through audio-visual context awareness. Extensive experiments show that their unified framework consistently outperforms cascaded baselines, demonstrating the effectiveness of the proposed task, dataset, and methodology.

**Strengths:**

While research on Audio-Visual Speech Recognition (AVSR) has been extensive, benchmarks for Audio-Visual Speech Understanding (AVSU) remain scarce. This work addresses that gap by providing a benchmark specifically designed to support the AVSU task.

The authors constructed an end-to-end learning framework for audio-visual speech understanding using a curated question–answer dataset tailored for speech comprehension. They further demonstrate that the proposed model effectively understands audio-visual speech and achieves notable performance improvements through various training and optimization techniques.

**Weaknesses:**

Since the original dataset used in this work, LRS3, is derived from TED talks, it exhibits dataset bias — the data mainly come from limited and controlled environments (e.g., speakers on stage, minimal emotional variation, and consistent speaking settings), making it less representative of general audio-visual speech understanding scenarios.

The baselines presented in Table 2 are insufficient. In particular, for the concatenated system, further comparison with other strong baselines such as GPT-4o, Gemini 2.5 Pro, or Video-SALMONN is needed to verify whether the proposed method indeed outperforms them.

* Minor weaknesses:
    * In Figure 2, the sections titled Lip-enhanced Perception, Question Analysis, Cues Explore, and Formal Conclusion are labeled in a way that may cause confusion with the framework itself.

    * Line 212: There are two bullet points and a mismatched table reference.

**Questions:**

* How was the hit rate in Figure 4 measured? Could you clarify how a “hit” at each stage is specifically defined?

* Although the paper states that the RL-based training approach contributes most significantly to performance improvement, the ablation study results do not seem to strongly support this claim. Is there any further analysis or explanation for this discrepancy?

* It appears that the dataset has not yet been released. Do the authors have plans to make the dataset publicly available in the future, possibly along with a demo?

---

### Official Review · Reviewer_h3MV · 2025-11-02

**Soundness:** 3
**Presentation:** 3
**Contribution:** 3
**Rating:** 4
**Confidence:** 4

**Summary:**

This paper addresses the audiovisual speech understanding (AVSU) task, extending conventional audiovisual speech recognition (AVSR) by jointly leveraging speech recognition and lip-reading for question answering. The authors construct a new dataset by generating question–answer pairs from transcriptions, automatically for the training set and semi-automatically for the evaluation set, with human validation incorporated in the latter. This dataset represents a valuable contribution, as the inclusion of human-in-the-loop validation ensures higher quality and reliability of evaluation data. Furthermore, the paper proposes a novel end-to-end AVSU framework employing multi-stage Chain-of-Thought (CoT) training, inspired by recent advances in reasoning-based learning. Experimental results demonstrate consistent performance improvements over cascade-based systems, supported by comprehensive ablation studies highlighting the effectiveness of the proposed CoT training strategy.

**Strengths:**

- The paper introduces a novel benchmark for audiovisual speech understanding (AVSU), which includes both automatically generated training data and human-validated evaluation data, providing a valuable resource for future research.
- It proposes a new end-to-end AVSU framework based on multi-stage Chain-of-Thought (CoT) training (four fine-tuning stages in total), effectively incorporating recent advances in reasoning-based learning from large language models (LLMs).
- The experimental results clearly demonstrate the effectiveness of the proposed end-to-end approach compared with cascade systems, supported by well-designed ablation studies that validate the contribution of each CoT training stage.

**Weaknesses:**

* **Terminology and scope:** The paper primarily addresses **question answering (QA)** and **intention understanding**, rather than the broader **spoken language understanding (SLU)** domain. SLU typically encompasses a wider range of tasks, including **localization/retrieval**, **semantic parsing**, and **entity extraction**. I recommend the authors review the **SLUE benchmark** (Shon et al., *“SLUE Phase-2: A Benchmark Suite of Diverse Spoken Language Understanding Tasks,”* ACL 2023) to better position their work. Given this focus, it may be more appropriate to rename the task as **AVSQA (Audiovisual Spoken Question Answering)** to avoid confusion.

* **Training complexity:** The proposed **four-stage training pipeline**, including the final **reinforcement learning step**, introduces considerable complexity. This raises concerns about reproducibility and accessibility. The authors should mitigate this issue by **fully disclosing the training procedures**, including **data preparation details** and **hyperparameter configurations** for each stage.

* **Task limitation:** As acknowledged in Section 6, the current system essentially combines **noise-robust speech recognition** with a **language-based QA module using lip reading**. The QA component itself does not leverage **visual context for reasoning**, limiting the contribution to true audiovisual understanding.

* **Evaluation metrics:** The paper relies solely on the **BERTScore** for evaluation. To provide a more comprehensive and reliable assessment, additional metrics such as **ROUGE**, **Exact Match (EM)**, **F1**, or **LLM-based evaluation** methods should be included, particularly given the open-ended nature of the QA task.

**Questions:**

* I may have overlooked it, but **does the paper include separate results for audio-only and visual-only AVSR performance**? Such comparisons are typically reported to illustrate the contribution of each modality in the AVSR experiments.
* Similarly, could you **report the QA performance using ground-truth ASR transcripts** to establish an upper bound for the text-only modality?
* It is not clear whether the **dataset created in this work will be publicly released**. Please clarify the availability and licensing of the data.
* In **Section 4.2.1**, please elaborate on the **training method** used — is it based on **embedding alignment** or **downstream fine-tuning**?
* In **Section 4.2.2**, please specify **which task** is being used for task-specific training.
* The **font size in Figure 4** is too small to read comfortably; please consider increasing it for clarity.

---

### Official Review · Reviewer_vgHy · 2025-11-07

**Soundness:** 1
**Presentation:** 3
**Contribution:** 1
**Rating:** 2
**Confidence:** 3

**Summary:**

The paper presents a new audio-visual understanding benchmark with an emphasis on liptracks, noisy speech and needing to integrate audio-visual modalities to improve understanding in these adverse conditions. The authors introduce a VSPEECH-R1 multimodal LLM that incorporate various off the shelf components (Whisper, AV-Hubert, Qwen) and how to train them together to yield an improved system over cascade.

**Strengths:**

The introduction of a AVSU-Bench evaluation for measuring LLM understand in a difficult environment that requires strong alignment of audio and visual modalities to achieve good performance. It is predictive in that it demonstrates the improvement a jointly trained model can have over cascaded. The paper also provides a recipe for training an LLM with off the shelf Whisper, AV-Hubert encoders with pre-trained Qwen LLM to yield a new VSPEECH-R1 multimodal LLM that surpasses the cascaded system.

**Weaknesses:**

A major weakness in the paper is that there are already quite a few audio-visual/multimodal tasks available, e.g.
* MMAU-Pro A Challenging and Comprehensive Benchmark for Holistic Evaluation of Audio General Intelligence https://arxiv.org/pdf/2508.13992
* MINERVA: Evaluating Complex Video Reasoning   https://arxiv.org/pdf/2505.00681
* https://sightsound.org/papers/2024/Li_AVQA-CoT_When_CoT_Meets_Question_Answering_in_Audio-Visual_Scenarios.pdf
* UniAV: Unified Audio-Visual Perception for Multi-Task Video Event Localization  https://arxiv.org/abs/2404.03179
however, the paper doesn't compare this task with these and others in the field.

Another weakness is with the statement "We further propose VSPEECH-R1, the first-ever end-to-end multimodal large language model tailored for AVSU." This statement is somewhat debatable since there are many other preceding multimodal LLMs. However, there are definitely recent general-purpose LLMs that can handle multimodal input, so it would be valuable to have comparisons with them, e.g. ChatGPT and Gemini.

Further, while VSPEECH-R1 shows gains over the cascaded system on AVSU-bench, there could be comparisons on the existing multimodal audio understanding benchmarks above.

Reproducibility is also limited in that no scripts/code is provided to more easily enable re-generating results.

**Questions:**

How does the AVSU-bench dataset correlate with other multimodal understanding datasets, eg MMAU-Pro, MINERVA, AVQA-CoT, UniAV?

How does the VSPEECH-R1 model compare with other LLMs that accept audio-video input such as ChatGPT and/or Gemini?

Do you have experiments of  VSPEECH-R1 vs the cascade approach on MMAU-Pro, MINERVA, AVQA-CoT, UniAV?

---

### Note · Authors · 2025-12-02

I have read and agree with the venue's withdrawal policy on behalf of myself and my co-authors.